# West Nile Virus Meningoencephalitis—A Consideration for Earlier Investigation

**DOI:** 10.3390/reports7020023

**Published:** 2024-03-27

**Authors:** David Burns, Zachary Vinton, Min Kyung Chung, Johnny Cheng

**Affiliations:** 1College of Osteopathic Medicine—Colorado Campus, Rocky Vista University, Englewood, CO 80112, USA; zachary.vinton@rvu.edu (Z.V.); jcheng@rvu.edu (J.C.); 2College of Osteopathic Medicine—Utah Campus, Rocky Vista University, Ivins, UT 84738, USA

**Keywords:** West Nile Virus, flavivirus, arbovirus, mosquito, meningoencephalitis, altered mental status, neuroinvasive disease

## Abstract

West Nile Virus (WNV) is an arbovirus endemic to many countries and has caused over 56,000 cases, with 2776 deaths in the U.S. from 1999 to 2022. WNV occurs most often in the fall, typically affecting elderly populations in states like Nebraska and Arizona. Currently, supportive care is the only management for WNV. Our case is a female patient in her mid-70s in an intermountain state who presented in the fall with WNV meningoencephalitis and experienced a delay in care due to the unique clinical presentation. This demonstrates the importance of early inclusion of WNV in the differential for altered mental status, especially with WNV risk factors, and expedition of supportive care. Doing so could potentially reduce antibiotic duration and hospital costs.

## 1. Introduction

West Nile Virus (WNV) is a mosquito-borne flavivirus first found in Uganda in 1937 and later introduced to the United States in 1999 [1,2]. Upon its incursion into the United States, it was thought to be responsible for 62 cases of neuroinvasive disease and seven deaths, which eventually led to thousands of infections throughout New York City [3]. Between 1999 and 2022, there were nearly 57,000 cases, with 25,769 requiring hospitalization, resulting in 2773 deaths [1]. Most cases have been found in states like Colorado, California, New Mexico, Nebraska, Illinois, and Texas, but the Centers for Disease Control and Prevention (CDC) notes that cases have been found in nearly every state [1]. Although WNV often infects humans, mosquito vectors may also transmit disease to birds, dogs, horses, and many other animals [3]. With changes to the climate including increased temperatures, milder winters, droughts, increased rainfall, and changing bird migratory and breeding patterns, case numbers are expected to increase due to increased survivability and reproductions of vectors [4].

Although many of these infections result in symptoms such as a minor fever, headache, malaise, and myalgias, they may occasionally lead to neuroinvasive disease. Neuroinvasive disease may manifest as meningitis, encephalitis, and/or acute flaccid paralysis. Between 2009 and 2018, the CDC analyzed 21,869 cases of WNV and determined that 59% were neuroinvasive cases including encephalitis and meningitis. Case fatality ratios for WNV encephalitis reached 14% and were found to be higher in patients over 70 years old [5]. The current treatment protocol mainly includes symptomatic management, as there is no antiviral treatment for WNV [6].

We present a case report of a patient in her mid-70s in an intermountain state with neuroinvasive WNV, namely meningoencephalitis, confirmed with WNV IgM in her cerebrospinal fluid (CSF). Due to the increased incidence of WNV infection in the U.S. and a high case fatality ratio, it is necessary for earlier consideration in differentials for altered mental status (AMS) with risk factors like increased age, immunocompromised states, the fall season, in geographic areas with high case numbers [7,8].

The purpose of this case report, as well as the methods that were used to conduct the literature search, are available in Appendix A.

## 2. Detailed Case Description

The patient is an elderly female with a past medical history of breast cancer in remission, remote asthma, macular degeneration, and Chiari malformation who was brought to the emergency department (ED) by her family with altered mental status (AMS) with a last known normal of two days previously. The day prior to admission, the patient reported to her daughter that she felt dizzy, confused, and febrile, with left eye swelling and pain. She was brought to the ED of one hospital, was reported to have normal vitals, received a routine complete blood count (CBC), basic metabolic panel (BMP), and urinalysis, which all were normal, and was discharged back home. Patient records including her lab work and diagnosis on discharge were unable to be obtained from this ED and her history was reported by the patient’s family.

Overnight, the patient was found by her daughter to be increasingly altered with urinary incontinence, so an ambulance was called. En route, it was reported that she was hypoxic at 80% on room air, was given 2 L of supplemental oxygen, and had a brief run of ventricular tachycardia, which resolved on arrival at the ED. On arrival, the patient was agitated and confused, severely septic with a temperature of 103.8 °F (Table 1), tachypneic, and had a lactate of 2.1. Physical exam findings were limited due to agitation, but it was reported that she exhibited “twitching of her bilateral lower extremities” and sluggish pupils, as shown in Table 1. CBC revealed leukocytosis with neutrophilic predominance (Table 2). Other notable labs included hyponatremia at 129 and elevated C-reactive protein (Table 3) with normal urinalysis and venous blood gas. Chest X-ray was normal with no acute cardiopulmonary process. Although the patient was unable to respond to questions, her family reported that the patient had no sick contacts, recent travel history, or animal exposure, but she could have been exposed to mosquitoes or other insects as she had been frequenting her porch and backyard during sunset. The patient lives alone in a suburban area with no exposure to farmland, chicken coops, or stables.

The differential included etiologies such as meningitis, stroke, toxic-metabolic encephalopathy, electrolyte derangement, and urinary tract infection (UTI). Due to her severe sepsis and suspected meningitis, fluids, empiric antibiotics, and antivirals consisting of vancomycin, ceftriaxone, meropenem, and acyclovir were initiated, and a lumbar puncture (LP) was ordered. Subsequently, the patient was admitted from the ED to the intensive care unit (ICU) on the same day. Although initially unsuccessful due to agitation on the first day of hospitalization, the next LP attempt four hours later was successful and revealed aseptic meningitis with over 1000 red blood cells (RBC), as shown in Table 4. It was noted that this could have been due to a traumatic tap. Gram staining was ordered and reported to be negative.

Cranial imaging including computed tomography (Appendix A) and magnetic resonance imaging (Appendix A) showed no acute intracranial findings such as masses or hemorrhages with a limited possibility of orbital infection progressing to the brain. An electroencephalogram (EEG) revealed diffuse slowing suggestive of encephalopathy. It was reported that the patient experienced a seizure during her hospitalization, but this was not confirmed by any other witness or seen on EEG. However, in order to be diagnostic, the EEG would have needed to be conducted during the seizure episode. Blood cultures for bacteria were negative. In order to rule out potential viral etiologies leading to the patient’s altered state, polymerase chain reaction (PCR) testing was ordered and revealed negative results for Herpes Simplex Virus (HSV), Varicella Zoster Virus (VZV), and WNV within seven days of admission (Table 5).

Due to high clinical suspicion for WNV specifically, an enzyme-linked immunosorbent assay (ELISA) for CSF IgM and IgG WNV antibodies was also ordered. On the seventh day of admission, the assay result yielded positive WNV IgM, confirming the diagnosis of WNV infection (Table 6).

Subsequently, all broad-spectrum antibiotics and antivirals were discontinued, and general supportive care was continued. The patient’s mentation slowly improved throughout her hospital course with amelioration of her neurologic symptoms except for her residual upper extremity tremors. She was later discharged with levetiracetam for the initial presentation of seizure with referrals to neurology, physical therapy, and occupational therapy.

## 3. Discussion

The risk factors that lead to the development of neuroinvasive disease are not well-defined; however, Bode et al. emphasizes that a greater incidence of severe neuroinvasive WNV is seen in elderly patients and patients who abuse alcohol and/or have diabetes mellitus [8]. Some of the risk factors demonstrated by our patient include her age and the general geographic location. Although neuroinvasive WNV is a rare complication occurring in less than 1% of infected cases [7], the various clinical features need to be recognized and further explored as part of a complete differential.

In patients infected with WNV, neurological complications can arise with clinical features of encephalitis, meningitis, and/or acute flaccid paralysis [2]. In particular, the characteristic symptoms of WNV encephalitis include depressed or altered level of consciousness, lethargy, personality changes lasting more than or equal to 24 h, and/or seizures. Our patient initially presented with neurological complications, including an altered level of consciousness with confusion and dizziness, and a potential seizure. Meningitis may present with nuchal rigidity, headache, photophobia, and/or phonophobia [2,7,9]. Additional laboratory values of importance include CSF pleocytosis and peripheral leukocytosis [2,7,9]. There are also a variety of movement disorders that can be seen with neuroinvasive WNV, including, but not limited to, tremors, opsoclonus-myoclonus, and Parkinsonism [2,9,10,11,12]. Our patient’s altered mental status and level of consciousness slowly improved throughout her hospitalization with near complete resolution prior to discharge, but her upper extremity tremors remained until discharge.

Although case reports involving neuroinvasive WNV in the U.S. are limited, Manusov et al. presented two cases involving AMS, vision changes, meningeal signs with sepsis, and vague symptomatology confounded by chronic conditions such as cord compression and opioid use [13]. Similarly, our patient had nonspecific symptoms such as AMS and fever, but also had unique findings of eye pain and urinary incontinence. Given her normal laboratory workup from her initial ED visit, her LP was ultimately delayed which led to a lowered clinical suspicion for WNV. Although our patient eventually met sepsis criteria, she lacked visual changes or meningeal signs, which complicated the differential diagnoses being considered. Moreover, it is unclear whether her unilateral ocular symptoms were related to her WNV infection. Subsequent normal imaging (MRA/MRI Brain) did not suggest orbital infection. Rousseau et al. demonstrates numerous ocular findings, including linear chorioretinitis, which is most suggestive of neuroinvasive WNV, as well as other inflammatory findings such as iritis and uveitis [14]. Our patient’s ocular symptoms may have been correlated with her infection; however, additional prompt ophthalmologic evaluation was limited, most likely due to escalation of sepsis protocol.

Our patient’s atypical presentation continued to obscure the clinical picture with a brief episode of ventricular tachycardia, which resolved upon her second visit to the ED, along with the aforementioned persistent bilateral lower-extremity twitching. Cardiac arrhythmias, although uncommon, have been observed particularly in patients diagnosed with West Nile Encephalitis [8]. Furthermore, as there are multiple different neurologic manifestations of neuroinvasive disease, tremors predominantly manifest in the upper extremities of patients rather than the lower extremities [9,15]. As such, these vague findings in our patient may or may not have been associated with neuroinvasive disease, yet they indicate the importance for considering a broad differential that includes WNV.

As our patient’s condition continued to progress with prolonged AMS and signs of encephalopathy, the hospital team increasingly suspected WNV, especially given its increased incidence in the intermountain state. Other infectious etiologies including bacterial and viral infections could not be immediately excluded, so an antibacterial and antiviral regimen was chosen and administered prior to the first LP attempt because of the patient’s AMS. Vancomycin and ceftriaxone were chosen for standard empiric coverage, with meropenem added for coverage of Listeria and intracellular bacteria. Ampicillin would have been used in place of meropenem, but the patient had a reported penicillin allergy. An abnormal LP with elevated total nucleated cells, elevated protein, normal glucose, and no growth on bacterial gram staining suggested aseptic meningitis and prompted further evaluation with an EEG, viral PCR, and antibody testing. Although her increased red blood cells found in the CSF could have suggested an etiology such as herpetic infection, the degree to which they were consistently elevated suggested a traumatic tap.

In consideration of the patient’s potential seizure, the differential included electrolyte abnormalities, trauma, cerebrovascular disease, brain tumors, abscesses, or neurodegenerative disease. Many of these etiologies were ruled out by cranial imaging. However, on admission, the patient exhibited moderate euvolemic hypoosmolar hyponatremia, which could have been a contributor. Sodium levels and osmolality began to correct over the next few days following the administration of normal saline, and there were no further reports of seizure-like activity. This made etiologies such as SIADH less likely, and poor intake or dehydration more likely. Her EEG, although not conducted around the time of the potential seizure, was negative for epileptiform abnormalities and non-specific for diffuse slowing, which is also seen in many encephalopathies caused by different etiologies. Gandelman-Marton et al. highlights similarities in eight patients with WNV-associated meningitis or meningoencephalitis with a prolonged clinical course that demonstrated predominantly anterior generalized, continuous slowing [16].

Given the laboratory findings suggestive of aseptic meningitis on LP, various infectious etiologies were considered for the patient’s presentation including WNV, HSV, and VZV. Our patient underwent PCR testing for these viruses, as well as an ELISA assay for WNV per CDC guidelines [16]. It is advised by the CDC to conduct multiple serologic tests simultaneously because the likelihood of detecting WNV with PCR alone is low [17]. Other tests that can be run with PCR include ELISA and immunohistochemistry. Although ELISA with positive results for WNV in serum or CSF is usually sufficient for diagnosis of WNV, plaque-reduction neutralization tests (PRNT) may be used to help confirm diagnosis if there is a concern for cross-reaction using the ELISA assay with another flavivirus such as St. Louis Encephalitis, Dengue, or Yellow Fever [17]. Although these flaviviruses were not tested for directly, the patient lived in an intermountain state with no recent travel history, which made infections from these other viruses unlikely. Our patient’s ELISA assay results were negative for WNV IgG antibodies and positive for WNV IgM antibodies in her CSF by day 7 of her hospitalization. The timing of these findings is consistent with the majority of patients with WNV encephalitis or meningitis who demonstrate positive IgM antibodies by the eighth day of symptom onset [18]. The patient’s negative WNV IgG and positive IgM results highlight the chronology of WNV infection. IgG usually rises a few days after IgM antibodies are detected and suggest chronic infection [17]. Positive IgM with negative IgG antibodies suggest an acute infection. These serologic findings along with her demographics strongly suggest that her clinical presentation was due to an acute neuroinvasive WNV infection. However, given that only her IgM was positive, the possibility of a false positive result cannot be ruled out and should be taken into consideration.

While there is strong evidence in this case that points to WNV encephalitis diagnosis, there is a limitation to our study. This patient’s WNV encephalitis presentation was unique not only because it is rare in incidence, but also because the patient showed signs of another infection that could have contributed to their neurological presentation. The patient was reported by her family to have experienced urinary tract infection symptoms including frequency several days prior to her admission. It is important to note that UTI may present with a normal urinalysis. When UTI is left untreated or is undertreated, it can present with altered mental status, also known as delirium [19]. Since this patient may have been experiencing UTI symptoms and was quickly discharged without management, it is possible that her UTI may have contributed to encephalopathy. WNV may have coincidentally compounded on top of her existing issues. Multiple days on empiric antibiotics could have cleared her infection and brought her mental status almost to her baseline, thus making it challenging to conclude that the cause of encephalopathy and her neurologic symptoms are solely due to neuroinvasive WNV.

## 4. Conclusions

Since this case report highlights an atypical presentation of neuroinvasive WNV, there is a limit to the generalizability of this case to others. However, given the complexity of neuroinvasive WNV, these findings supplement the existing literature by exemplifying the possible clinical presentations of AMS that providers need to consider when generating their list of possible differentials. Although there are more typical features of neuroinvasive disease that can be expected, this is an important case of WNV meningoencephalitis that highlights the potential ambiguities in neurodegenerative disease, which ultimately delayed her diagnosis. When considering elderly patients that present during the fall season in the U.S. with altered mental status, WNV should be included early in the differential, and if clinically suspicious, should be promptly evaluated for with LP and diagnostic testing. If the clinical presentation is atypical such as our case, early admission for observation and supportive care may be necessary to hasten evaluation for neuroinvasive disease. Providers should keep these factors in mind as neuroinvasive WNV may present with different combinations of clinical features that evolve over time, making clinical diagnosis very difficult and postponing necessary patient care. Our purpose is to better inform providers of the varied presentations of WNV, as well as risk factors that should prompt early investigation in patients with AMS. We have learned from this case that severe manifestations of neuroinvasive disease may present suddenly, and that it is important to be aware of the variable or ambiguous symptoms that patients may present with. Diagnosis of WNV is dependent on obtaining CSF and antibody testing, which may take several days, so this must be done promptly in cases where etiologies such as electrolyte derangement, acute metabolic encephalopathy, and urinary tract infections are ruled out.

## Figures and Tables

**Table 1 reports-07-00023-t001:** Vital Signs and Physical Exam Findings in the Emergency Department.

Item	Patient
Pulse Oximetry (%)	95
Blood Pressure (mmHg)	155/74
Pulse (Beats per minute)	101
Respiratory Rate (Breaths per minute)	20
Temperature (°C)	39.9
Physical Exam	Findings
General	Altered, no acute distress
Skin	Normal color, warm, dry and intact, no lesions or rashes noted
Head/Eyes	Normocephalic, EOMI, PERRLA, normal conjunctiva/sclera
Ears	External auditory canals and tympanic membranes clear, hearing grossly intact
Nose	No nasal discharge
Mouth	Moist mucous membranes, no pharyngeal erythema
Neck	Supple, no meningismus, full range of motion
Breast	Bilateral mastectomy
Cardiac	Normal S1 and S2, no murmur, rubs, or gallops, rhythm is regular, no peripheral edema
Lungs	Clear to auscultation without rales, rhonchi, wheezes, or diminished breath sounds
Abdomen	Normoactive bowel sounds x4, soft, nondistended, nontender, no guarding or rebound, no masses
Musculoskeletal	Full range of motion in upper and lower extremities, no joint erythema or tenderness, gait not tested due to patient condition
Neurological	Alert and oriented x1, garbled speech, no sensory or motor deficits, patient twitching bilateral lower extremities, GCS 15

**Table 2 reports-07-00023-t002:** Initial CBC Upon Admission.

Item	Patient	Reference Range with Units
WBC	12.98	3.80–10.80 10^9^/L
Neutrophils	10.15	1.41–6.69 10^9^/L
Lymphocytes	2.03	0.85–3.90 10^9^/L
Monocytes	0.70	0.20–0.95 10^9^/L
Eosinophils	0.01	0.02–0.50 10^9^/L
Basophils	0.03	0.01–0.20 10^9^/L
Nucleated RBC	0.00	0.00–0.00 10^9^/L
RBC	4.59	3.96–5.31 10^6^/mcL
HgB	13.4	12–16.3 g/dL
HCT	39.9	36.8–49.5%
Platelet	237	140–400 10^9^/L

**Table 3 reports-07-00023-t003:** Initial Serum Chemistry Upon Admission.

Item	Patient	Reference Range with Units
Sodium	129	136–145 mmol/L
Potassium	4.0	3.5–5.1 mmol/L
Chloride	98	98–107 mmol/L
Carbon dioxide	24	21–32 mmol/L
Anion gap	7	3–15 mmol/L
BUN	8	7–18 mg/dL
Creatinine	0.88	0.55–1.02 mg/dL
Calcium	8.6	8.5–10.1 mg/dL
C-reactive protein	0.80	<0.29 mg/dL
Lactate	2.1	0.4–2.0 mmol/L
Random Glucose	114	74–106 mg/dL

**Table 4 reports-07-00023-t004:** Cerebrospinal Fluid Analysis.

Item	Reference Range with Units	Tube 1	Tube 2	Tube 3
Appearance	Clear	Hazy	-	Hazy
Color	Colorless	Pink	-	Xanthochromic
RBC	0–10 mcL	1550	-	1125
Total nucleated cell	0–5 mm^3^	141	-	93
Mononuclear cells	15–45%	14	-	24
Neutrophils	0–5%	55	-	47
Lymphocytes	40–80%	31	-	29
Glucose	40–75 mg/dL	-	50	-
Total protein	8–32 mg/dL	-	97	-

**Table 5 reports-07-00023-t005:** Viral PCR Results (Serum).

Item	Patient	Reference
HSV	Negative	Negative
VZV	Negative	Negative
WNV	Negative	Negative

**Table 6 reports-07-00023-t006:** WNV Antibody ELISA Results (CSF).

Item	Patient	Reference Range
IgM	3.45	0.00–0.90
IgG	-	0.00–1.30

## Data Availability

The original contributions presented in the study are included in the article/Appendix A, further inquiries can be directed to the corresponding author.

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
