# Peer review of "West Nile Virus Meningoencephalitis—A Consideration for Earlier Investigation"

_reports, 2024, doi:10.3390/reports7020023_

Round 1

Reviewer 1 Report

Comments and Suggestions for Authors

Thank you for the opportunity to review this manuscript. It is an interesting topic and an important abroviral infection.

Here are some suggestions to consider.

A detailed methodology must be included in this Case Report. 

Provide the age of the patient and her past medical history in particular was she on any medication? Was there prior hyponatremic events? Is it SIADH? prior UTI?

Was there any recent travel history? any animal exposure? consumption of probable food source of infection?

Her residence is it in an urban city or rural area with stables or poultry coops?

From her initial visit to the ED, what was the diagnosis at discharge for her presenting complaints? If possible, providing the serial hematological and biochemical profiles as Supplementary Materials will be useful for the readers to understand the clinical picture. 

Include a detailed physical examination. What was her GCS at presentation? Please include her vital signs (Body temperature, blood pressure, pulse, respiratory rate, also oxygen saturation). It appears she might be in the state of shock with elevated lactate levels. What was the capillary refill time? Was the presence of rash noticed anywhere? Please provide the serum glucose level at presentation. In addition, please add a complete neurological examination and findings. Especially since this case is described as an acute infection of West Nile virus infection and the severe manifestation  of WNV is mostly described as neurotropic.

What about her hearing?

Was there any non-invasive measures to estimate any depletion of intravascular volume? CXR taken?

There was a documented episode of seizure during this acute event. Was CK or CKMB sent? Both hyponatremia, arrhythmia and infection can lead to seizures. Reading the description of the patient exhibiting a confused toxic state, in my opinion seems like meningoencephalitis. 

Is human hyperimmune immunoglobulins an option to administer if available? or was steroids administered? 

If possible, please provide rationale for the antimicrobial usage in this case. Additionally, which drug was given to cover for possible intracellular pathogens?

To avoid any confusion, please state if spinal tap was traumatic leading to the increased red blood cells observed, Also was any gram staining performed. For the molecularly CSF analysis please provide the details of all pathogens screened. 

Can we exclude cross-reaction to other neurotropic flaviviruses? 

Include the cranial imaging in Supplementary findings. Especially of areas involved during infection and findings commonly seen with meningoencephalitis. You may report as these findings were not present and provide selected slides as Supplementary Materials.

Was an echocardiography and blood cultures performed?

What was the workup and management of hyponatremia?

In the present version, the Introduction appears to be a single paragraph. Please consider breaking it down to multiple paragraphs and expanding further. As WNV infection is geographically limited to certain regions, it would be important to provide sufficient background in the Introduction to enhance readers who are not familiar with WNV.

Reviewer 2 Report

Comments and Suggestions for Authors

In the current manuscript, the authors present a case report of WN, which was characterized by an unusual clinical presentation that complicated early diagnosis. Overall, the case is well described, and the reference list is updated.

I have a suggestion: the authors should report the outcome of the patient and possibly the follow-up, it would help in the understanding of the disease and its consequences. I suggest the authors be more precise in the geographical details – admission Hospitals, departments or Units of hospitalization, and rehabilitation centers.

Minor concerns:

ABSTRACT: Please remove references from the abstract, edit the sentence “ the importance of early inclusion of WNV in the differential for altered mental status,” and remove unneeded abbreviations such as AMS, which is used only once in the abstract.

- CASE DESCRIPTION: Please avoid the use of little-used abbreviations or specify them (for example, CBC, CRP, VBG); MRA is never defined; the sentence “ It was reported that the patient experienced a seizure during her hospitalization, but this was not confirmed on EEG” is confusing, since the patients could have experienced a seizure even with an EEG showing only a slowing activity unless the EEG was performed during the suspected seizure episode; please specify when the lumbar puncture was performed, considering the first failed attempt – within 24 – 48 – 72 hours?

- DISCUSSION: Did the patient experience a first seizure or recurrent seizures (it is unclear in the sentence “while her seizures soon resolved following admission.” How was the outcome? Did the patient experience cognitive consequences? Was she evaluated from a neuropsychological point of view?

Comments on the Quality of English Language

Minor editing

Round 2

Reviewer 1 Report

Comments and Suggestions for Authors

The manuscript has been improved, but there is an issue with the excessive number of tables. It is recommended to combine some of them for better organization and clarity.

For instance, consider merging Table 1 and Table 4 into a single table placed before the laboratory investigations section.

Please remove the '#' in Table 2 to enhance readability.

Correct Table 6 and provide the reference range for accurate interpretation.

Additionally, it is advisable to include a detailed methodology section as previously suggested. This will provide readers with a clear understanding of the research methods employed.

Reviewer 2 Report

Comments and Suggestions for Authors

The authors fully replied to my concerns

Author Response

Thank you so much for allowing us to reply to your comments and concerns. 

Round 3

Reviewer 1 Report

Comments and Suggestions for Authors

Thank you for revising the manuscript. I have no further comments.

Author Response

Thank you so much for reviewing our manuscript.